# A Comprehensive Study on Microstructure and Wear Behavior of Nano-WC Reinforced Ni60 Laser Coating on 17-4PH Stainless Steel

Jie Wang, Xiaoqiang Zhang *, Lei Qiao, Yue Zhao, Mengfei Ren, Tiaotiao Li and Ruifeng Li *

School of Materials Science and Engineering, Jiangsu University of Science and Technology, Zhenjiang 212000, China; wangjie98727@163.com (J.W.); qiaolei0306@just.edu.cn (L.Q.); zhaoyue90214@just.edu.cn (Y.Z.); rmf519778357@163.com (M.R.); litaotao@just.edu.cn (T.L.)
* Correspondence: 202100000022@just.edu.cn (X.Z.); li_ruifeng@just.edu.cn (R.L.)

**Abstract:** The surface of 17-4PH martensitic stainless steel was laser-cladded with Ni60 and Ni60+nano-WC composites and a comprehensive investigation was conducted of the microstructure and wear mechanism. The findings demonstrate that despite the added nano-WC particles being fused and dissolved during laser cladding, they still lead to a reduction in grain size and a decrease in crystallographic orientation strength. Furthermore, the dissolution of nano-WC makes the lamellar $M_{23}C_6$ carbides transform into a rod or block morphology, and leads to the CrB borides becoming finer and more evenly dispersed. This microstructural evolution resulted in a uniform increase in hardness and wear resistance, effectively preventing crack formation. When the nano-WC addition increased to 20 wt.%, there was a 27.12% increase in microhardness and an 85.19% decrease in volume wear rate compared to that of the pure Ni60 coating. Through analysis of the microstructure and topography of wear traces, it can be inferred that as the nano-WC addition increased from 0 wt.% up to 20 wt.%, there was a gradual transition from two-body abrasive wear to three-body abrasive wear, ultimately resulting in adherent wear.

**Keywords:** laser cladding; microstructure; wear mechanism; composite coating





## 1. Introduction

Laser cladding is an advanced modern surface modification technology that has been widely applied in aerospace, petrochemical, and other industries [1,2]. Compared with traditional surface modification processes such as electroplating [3], thermal spraying [4], physical vapor deposition [5], and arc welding [6], laser cladding has obvious advantages such as a smaller heat-affected zone, higher dense coating structure, and higher bonding strength. With the development of laser cladding, the materials used for it are gradually changed from simple to complex to obtain enhanced wear resistance, corrosion resistance, or other special properties.

In the field of preparation of nickel-based alloy coatings, adding high-melting-point, hard compounds such as borides, oxides, or carbides to self-fluxing alloy powders to obtain excellent composite coatings has become a popular research topic [7–10]. The particle size of these added compounds could be on the micron or nano scale. In particular, WC is one of the most commonly added hard compounds, and ball milling mixing with original nickel-based alloy powder is the primary adding method. Research shows that the amount of micron WC added has significant effects on the phase, microstructure, and properties of the resulting coatings. Li et al. [11] studied the microstructure of the cladding coating formed by WC/Ni60A composite powders with different micron WC mass fractions under optimized process parameters. It was found that when the mass fraction of the added WC increased from 20% to 50%, the cladding coating changed from having a large number of dendrites and W-rich carbides, to a $M_6C$, $M_{23}C_6$, and acicular structure. The microhardness

of the WC coating is significantly improved compared with the substrate. M.J. Tobar et al. [12] carried out the laser cladding experiment of NiCrBSi/Ni-clad WC mixed powder on stainless steel substrate and studied the effect of the volume fraction of WC particles on the microstructure and properties of the composite coating. The results show that when the volume fraction of WC is less than 50%, a uniform, dense, and crack-free composite coating can be obtained. A Ortiz et al. [13] found that due to the density difference between WC particles and the Ni-based matrix, the distribution of remaining WC is non-homogenous, and there are more un-melted WC particles near the substrate interface. A García et al. [14] reached a similar conclusion in their study of WC/NiCrBSi laser cladding; moreover, they found that when the actual WC content exceeds 27 wt.%, the increase in WC content because of the increase in carbide cannot continue to significantly improve the wear resistance.

To improve the uniformity of the microstructure and crack resistance of the coating, researchers carried out studies of nano-WC reinforced coatings based on micron WC particle reinforced coatings. Compared to micron particles, nanoparticles have the advantages of small size, large specific surface area, catalytic activity, and significant advantages in strength, toughness, plasticity, and corrosion resistance. Adding nano-WC can not only hinder the growth of dendrites but also promote heterogeneous nucleation to reduce grain size. The microstructure of the coating can be modified to enhance its wear resistance and corrosion resistance [15–18]. Additionally, the incorporation of nanoparticles facilitates improved interface contact, thereby reducing crack formation and defects, and ultimately enhancing adhesion between the coating and substrate [19]. However, compared to micron WC, nano-WC is more easily dissolved into the coating matrix during laser cladding; thus, it is difficult to retain enough undissolved WC. This phenomenon may lead to a reduction in hardness and wear resistance [15].

According to the aforementioned research reports, it is evident that the distribution and existence of WC particles in composite coatings have a significant effect on the microstructure and performance of the composite coatings. If the added WC is at the micron scale, the amount of WC remaining in the coating is high, but the distribution is not uniform, and too high a content of WC would cause the coating to crack. On the contrary, if the nanoscale WC is added, then the WC is likely to dissolve, resulting in a reduction in the remaining WC content in the obtained coating, but at this time, the microstructure is more uniform and the crack resistance is enhanced. However, the changes in microstructure and the resulting performances caused by the dissolution of nano-WC are still not clear for WC/Ni60 composite coatings, which hinders the development and application of high-quality and crack-free coatings. Therefore, this study aimed to comprehensively reveal the nano-WC dissolution behaviors and the resulting microstructure evolution and performance. The results will effectively facilitate industrial applications of nano-WC/Ni60 composite coatings.

## 2. Materials and Methods

### 2.1. Materials

17-4PH martensitic stainless steel with dimensions of $100 \times 100 \times 15$ mm$^3$ was used as the substrate for laser cladding, and was provided by Shanghai Yong Ye Industrial Co., Ltd. (Shanghai, China). The chemical compositions are exhibited in Table 1.

**Table 1.** Chemical compositions of laser cladding substrate (wt.%).

| Element | C | Si | Mn | P | S | Cr | Ni | Cu | Fe |
|---|---|---|---|---|---|---|---|---|---|
| 17-4PH | 0.07 | 1.00 | 1.00 | 0.035 | 0.03 | 16.5 | 4.20 | 4.50 | Bal |

The Ni60 powder utilized in the experiment was supplied by Shanghai Maoguo Nanotechnology Co., Ltd. (Shanghai, China). Table 2 shows its composition.

**Table 2.** Chemical compositions of Ni60 powder (wt.%).

| Element | C | B | Cr | Si | Fe | Ni |
|---------|------|-----|------|------|-----|-----|
| Ni60 | 0.62 | 3.3 | 18.5 | 0.06 | 5.0 | Bal |

The nano-WC powder used for preparing WC/Ni60 composite coating was provided by Jiangsu Xianfeng nanomaterials Technology Co., Ltd. (Nanjing, China). The original morphology is depicted in Figure 1a. The size of the nano-WC particles is predominantly below 100 nm. Figure 1b exhibits the phase constitution of nano-WC revealed by X-ray diffractometer patterns, which shows excellent purity. The nano-WC/Ni60 mixed powders with varying proportions were pre-prepared by electromagnetic stirring to ensure uniform mixing. Anhydrous ethanol was used as the stirring medium, the rotational speed was 600 r/min, and the stirring time was 1 h. After the electromagnetic stirring process, the composite was subjected to a drying period of 2 h at a temperature of 120 °C in a drying chamber. Figure 1c,d show the morphology of the obtained composite powders; the specific mixing ratio is presented in Table 3.

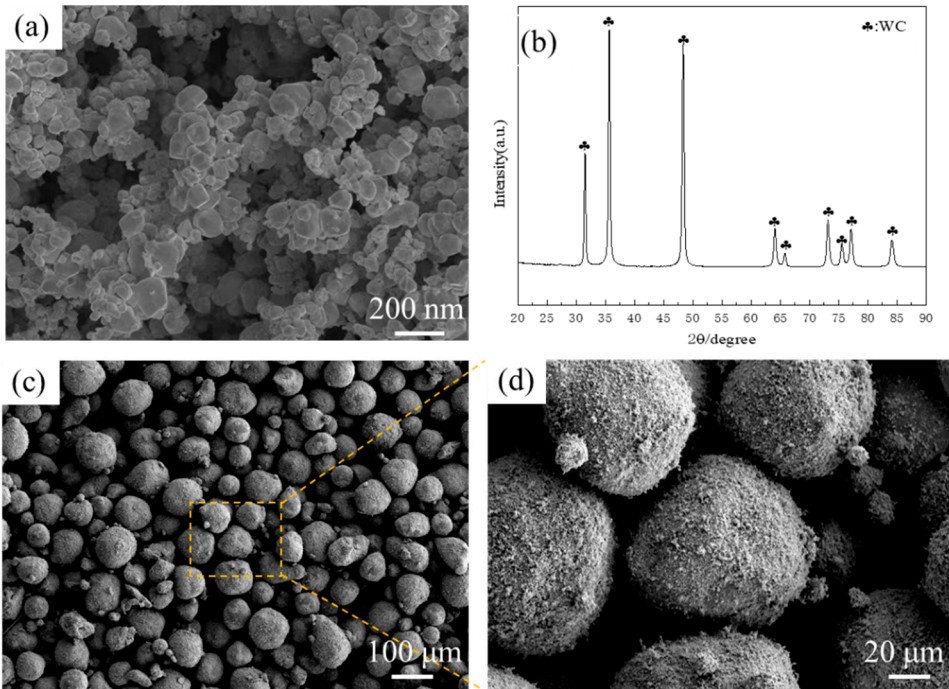

**Figure 1.** SEM morphology, XRD pattern of nano-WC, and SEM morphology of mixed powder: (**a**) shows the morphology of nano-WC; (**b**) is the XRD pattern of nano-WC; (**c**) is the morphology of nano-WC/Ni60 composite; (**d**) is the partial enlarged view of (**c**).

**Table 3.** The Ni60 and nano-WC proportion of the mixed powder (wt.%).

| Ni60 | 100 | 95 | 90 | 85 | 80 |
|---------|-----|----|----|----|----|
| Nano-WC | 0 | 5 | 10 | 15 | 20 |

### 2.2. Methods

Before the cladding, the substrates were ground and cleaned to remove the oxide and oil film, and they were preheated to 300 °C. The mixed nano-WC/Ni60 powders were evenly laid on the substrate with a scraper, and the layer thickness was 1 mm. The shielding gas was argon with purity of 99.99%.

After optimizing the process parameters during the preparation stage, the process parameters presented in Table 4 were selected for this study. Under this optimized process, the cladding layer exhibits consistent height, uniform size, and excellent integrity.

**Table 4.** Laser cladding parameters.

| Process Parameters | Value |
| --- | --- |
| Laser power (W) | 2000 |
| Scanning speed (mm/s) | 2 |
| Powder layer thickness (mm) | 1 |
| Laser spot size (mm$^2$) | $5 \times 5$ |
| The flow rate of shielding gas (L/min) | 10 |

The microstructures were observed by a Carl Zeiss field emission scanning electron microscope (SEM, Zeiss, Jena, Germany). JEM-2100F field emission transmission electron microscopy (TEM, JEOL Ltd., Tokyo, Japan) was employed to conduct detailed phase structure analysis. The TEM samples were ground to a thickness of 50 μm by sandpaper and punched into wafers measuring 3 mm in diameter before being thinned via a double electrolytic spray (10% $HClO_4$ + 90% $C_2H_5OH$). The cross-sectional microhardness of the coating was tested by an HXS-1000AC hardness tester (Shanghai Shangguang Optical Co., Ltd., Shanghai, China), with a weight of 200 g and 10 s dwell time. Four hardness values were tested at the same level. The wear resistance was evaluated through ball–block sliding friction tests at a speed of 350 rad/min for 1 h, with a test force of 20 N. The microtopography of the wear traces was observed by a LEXTOLS4000 confocal laser scanning microscope (Olympus, Tokyo, Japan).

## 3. Results

### 3.1. Microstructures

The microstructure observed by SEM and elemental distribution obtained by EDS of the pure Ni60 coating and the composite coating with nano-WC added to Ni60 are depicted in Figure 2. As can be seen from Figure 2a,b, the pure Ni60 is mainly composed of the matrix phase (location A), the lamellar eutectic structure located between dendrites (location B), and some scattered dark grey polygon blocks (location C). Analysis of the elemental distribution reveals that the lamellar eutectic structure exhibits high levels of Cr while being deficient in Ni. The microstructure and elemental distribution of the composite coating with 10 wt.% nano-WC added to Ni60 are presented in Figure 2c. Compared to Figure 2b, a notable distinction in Figure 2c is observed, wherein the lamellar eutectic structure situated amidst dendrites transforms into rod and block precipitates upon the addition of 10 wt.% nano-WC to Ni60, with these precipitates exhibiting an abundance of Cr and W. With the content of nano-WC increased to 20 wt.%, the microstructure and elemental distribution of the composite coating are depicted in Figure 2d. Compared to the microstructure of the sample to which 10 wt.% nano-WC is added shown in Figure 2c,d, the 20 wt.% sample exhibits a reduced presence of rod-shaped precipitates while showcasing an augmented occurrence of block-shaped precipitates between dendrites. Even when certain rod-shaped precipitates are observed, their dimensions appear coarser. Regarding elemental distribution, these precipitates are enriched in Cr and W, with higher W content detected in block-shaped precipitates compared to rod-shaped ones.

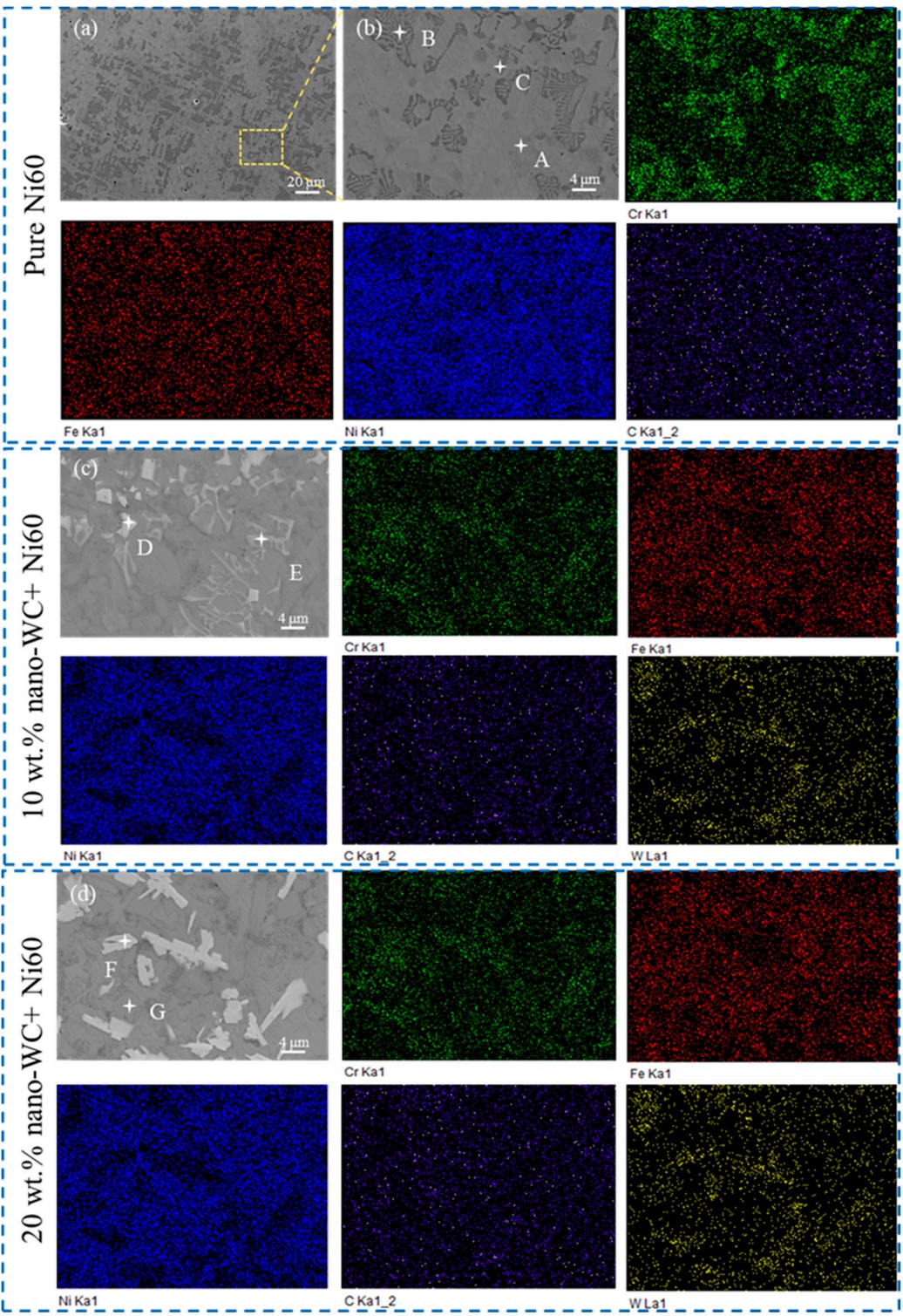

**Figure 2.** The microstructure observed by SEM and elements distribution obtained by EDS Ni60 coating: (**a**) pure Ni60; (**b**) an enlarged view of (**a**); (**c**) 10 wt.% nano-WC-added Ni60; (**d**) 20 wt.% nano-WC-added Ni60.

To compare the composition differences of the aforementioned microstructures, EDS analysis was conducted on the designated points. The results are presented in Table 5. It is important to note that light elements such as B and C were excluded from the analysis due to inherent inaccuracies in their EDS measurements. Point A is located in the matrix dendrite stem of the pure Ni60 coating, with a Ni content of 71.12 wt.% and Cr content of

13.92 wt.%. B contains 57.83 wt.% Ni and 28.94 wt.% Cr; this means the lamellar eutectic structure contains a Cr-enriched component phase, and the EBSD and TEM results below show that this phase is a carbide named $M_{23}C_6$, where M mainly represents the Cr element. The dark grey block phase marked by point C exhibits a higher Cr content (67.81 wt.%) compared to points A and B, which can be identified as CrB based on EBSD results. Points D, E, and F correspond to white grey block and rod phases observed in composite coatings with nano-WC additions at concentrations of 10 wt.% and 20 wt.%. Notably, these points exhibit equivalent levels of Cr content and can be also identified as $M_{23}C_6$ carbides rich in both Cr and W according to corroborating evidence from the EBSD analysis below. Moreover, it can be deduced that with the increase in Cr and W content, the rod $M_{23}C_6$ carbides tend to change into blocks.

**Table 5.** Composition analysis by EDS of the marked points (wt.%).

| Point | Cr | Si | Fe | Ni | W | Inferred Phase |
|-------|------|------|-------|-------|-------|----------------|
| A | 13.92 | 5.91 | 9.05 | 71.12 | — | Matrix $\gamma$ |
| B | 28.94 | 4.81 | 8.43 | 57.83 | — | $\gamma + M_{23}C_6$ eutectic |
| C | 67.81 | 1.18 | 6.89 | 24.12 | — | CrB |
| D | 29.66 | 7.53 | 4.82 | 25.19 | 32.81 | $M_{23}C_6$ |
| E | 33.07 | 6.01 | 8.66 | 37.60 | 14.98 | $M_{23}C_6$ |
| F | 36.85 | 5.83 | 7.26 | 21.33 | 37.37 | $M_{23}C_6$ |
| G | 11.05 | 0.63 | 11.46 | 76.01 | 0.85 | Matrix $\gamma$ |

To support the above inference about the phases, phases calibration and distribution detection were conducted by EBSD; the results shown in Figure 3. In particular, Figure 3a–c correspond to the laser coatings with 0 wt.% (pure Ni60), 10 wt.%, and 20 wt.% nano-WC addition, respectively. In these figures, the red phase represents $\gamma$, yellow corresponds to $Cr_{23}C_6$, and green indicates CrB phase. The morphology and distribution of these phases revealed by EBSD confirm the previously mentioned inference regarding the phase composition. It can be seen from Figure 3 that the increase in nano-WC addition leads to a gradual decrease in the content of matrix phase $\gamma$ in the coating from 79.9% to 71.7%, and its grains gradually change from dendritic to equiaxed. At the same time, the content of $M_{23}C_6$ type carbide gradually increases from 19.3% to 28.2%, and its shape gradually transforms from lamellar to rod and block. Furthermore, in the Ni60/nano-WC composite coating, the size of CrB particles is obviously reduced, and the distribution is more dispersed and uniform.

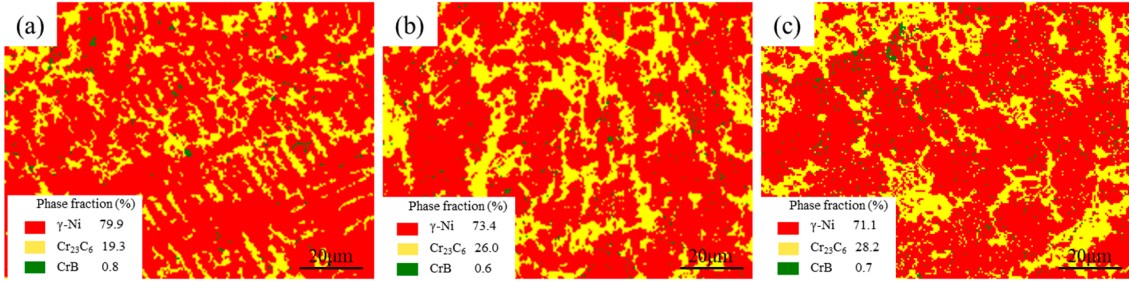

**Figure 3.** EBSD results of the coatings with different nano-WC contents: (**a**) 0 wt.%; (**b**) 10 wt.%; (**c**) 20 wt.%.

The pole figures of matrix $\gamma$ are presented in Figure 4, depicting the coatings with 0 wt.%, 10 wt.%, and 20% nano-WC additions as shown in Figure 4a, Figure 4b, and Figure 4c, respectively. The results indicate a decrease in the maximum texture densities of matrix $\gamma$ from 49.42 to 11.10 with an increasing amount of nano-WC added. The maximum of the texture densities serves as a credible index to describe the texture strength, and if this index exceeds 1, it signifies the presence of texture within the material. Therefore, it can be inferred that the incorporation of nano-WC reduces the strength of the coating's texture or

equivalently deteriorates the growth orientation of matrix γ grains. The increase in nano-WC addition leads to a reduction in the texture strength of γ crystal and an enhancement in grain orientation diversity within the same test area, thereby providing evidence for a decrease in grain size.

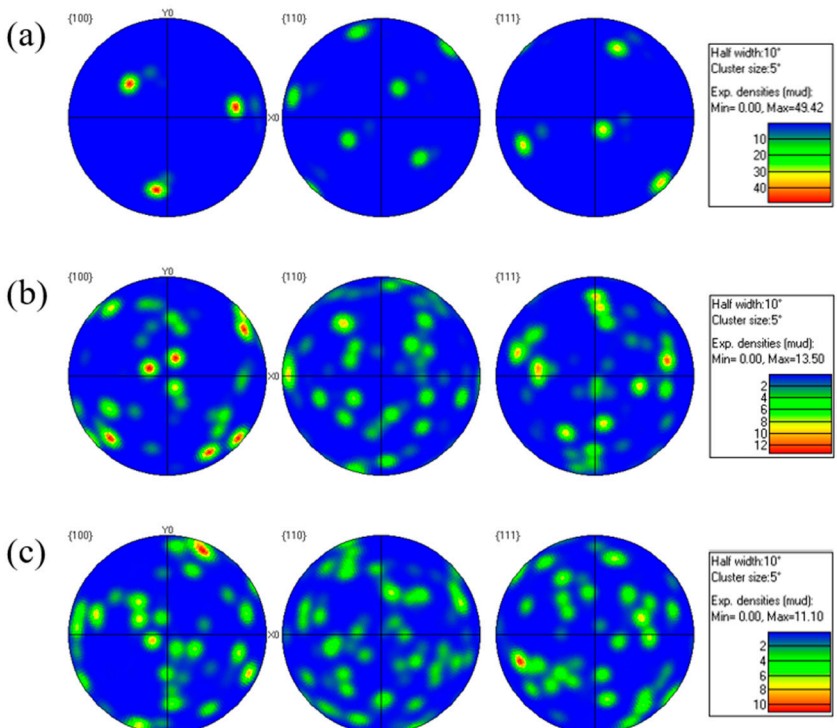

**Figure 4.** The pole figure of γ-Ni in different contents of nano-WC laser cladding coatings: (**a**) 0 wt.%; (**b**) 10 wt.%; (**c**) 20 wt.%.

For a more in-depth analysis of phase structure and composition when nano-WC is added, the microstructure of Ni60 + 20 wt.% nano-WC coating was analyzed by TEM; the results can be seen in Figure 5. Figure 5a,b show the bright field (BF) images at different magnification scales, Figure 5c,d show the selected area electron diffraction (SAED) results of the marked zones in Figure 5b, Figure 5e shows the zone selected for EDS analysis, and the EDS results are shown in Figure 5f–i. The SAED results further confirmed that the abundant rod- or block-shaped precipitates in the texture are $M_{23}C_6$ carbides, which are rich in Cr and W.

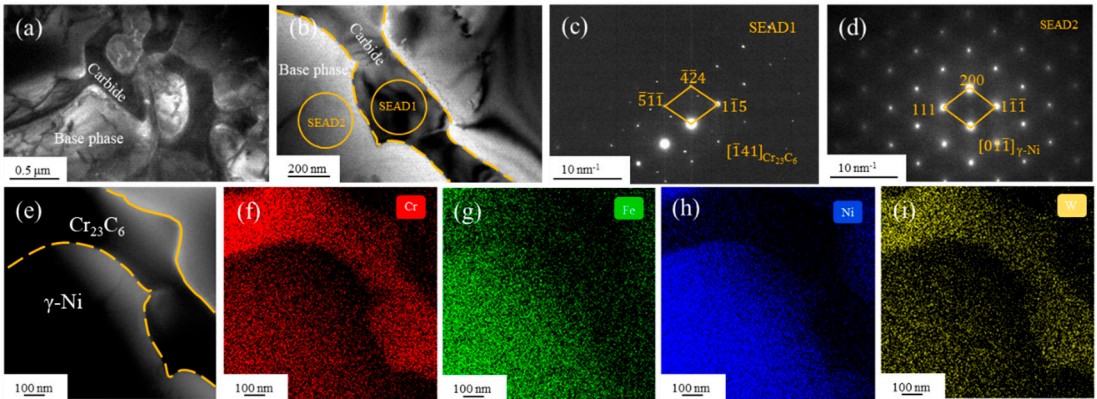

**Figure 5.** Phase analysis of $M_{23}C_6$ and γ-Ni: (**a,b**) BF images at different magnification scales; (**c**) SAED of $M_{23}C_6$; (**d**) SAED of γ-Ni; (**e–i**) the energy spectrum analysis.

The high-resolution analysis results at the interface between $M_{23}C_6$ and $\gamma$-Ni are shown in Figure 6. Figure 6a shows that there is a clear boundary between $M_{23}C_6$ and the $\gamma$-Ni interface, and the microstructure on both sides is different. Figure 6b shows the high-resolution amplification map at the interface between $M_{23}C_6$ and $\gamma$-Ni, and clear lattice fringes are observed at the interface and on both sides. The Fourier transform (FFT) of $\gamma$-Ni (Figure 6c) shows that the lattice fringes are mainly along the $[0\bar{1}1]$ $\gamma$-Ni crystal axis and the inverse Fourier transform (IFFT) shows that the crystal plane spacing $d_{(111)} = 0.2149$ nm. The FFT of $M_{23}C_6$ (Figure 6d) shows that the lattice fringes are mainly along the $[1\bar{2}1]$ $M_{23}C_6$ crystal axis and the IFFT shows that the interplanar spacing $d_{(333)} = 0.2066$ nm. It is found that there is a certain phase relationship between $M_{23}C_6$ and $\gamma$-Ni, in which $(111)_{\gamma\text{-Ni}} \| (333)_{Cr23C6}$. The interface strength between $M_{23}C_6$ and $\gamma$-Ni is related to the mismatch caused by the difference in lattice constants [13]. According to the results of the IFFT, the mismatch between $(111)_{\gamma\text{-Ni}}$ and $(333)_{M23C6}$ was calculated by the following formula:

$$\delta_d = \frac{|\,d\alpha - d\beta\,|}{\frac{d\alpha + d\beta}{2}} \tag{1}$$

where $d\alpha$ is the interplanar spacing of $(111)_{\gamma\text{-Ni}}$, and $d\beta$ is the interplanar spacing of $(333)_{M23C6}$. The calculated mismatch $\delta_d$ is 3.9%, which is less than 6%, indicating strong interface strength between $M_{23}C_6$ and $\gamma$-Ni. This is also one of the reasons why carbides can be precipitated in large quantities. Furthermore, this strong interface strength makes it difficult for the $M_{23}C_6$ to separate from the matrix under stress, which is conducive to reducing the cracking sensitivity of the cladding layer. Moreover, the lower mismatch $\delta_d$ means that the internal stress caused by $M_{23}C_6$ precipitation is lower, which is also conducive to crack suppression.

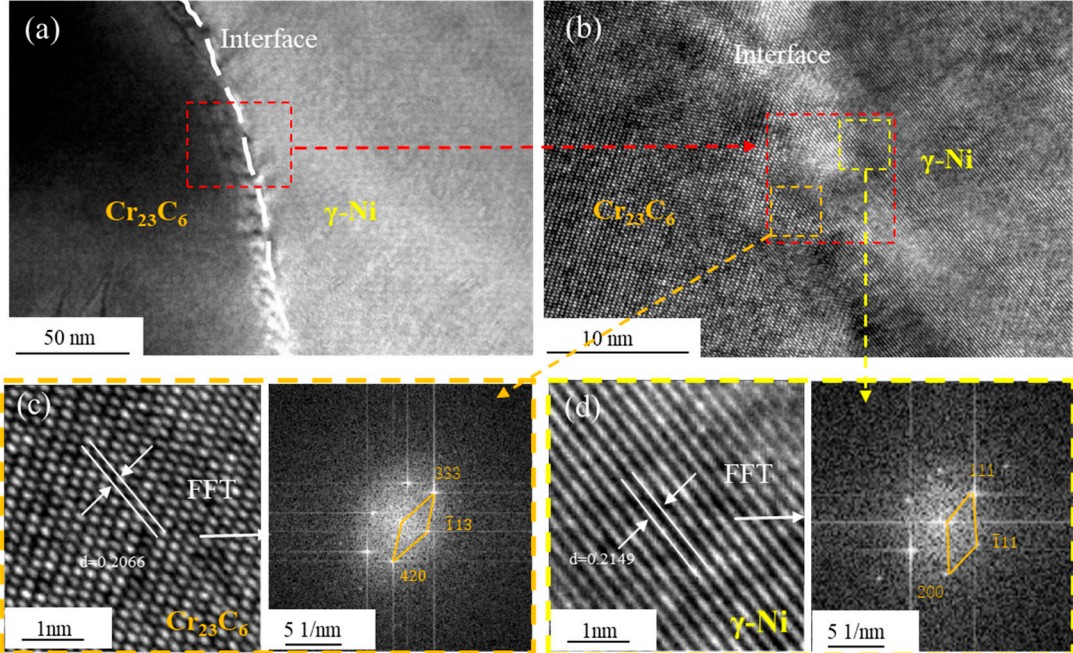

**Figure 6.** High-resolution analysis results: (**a**) $M_{23}C_6$ and $\gamma$-Ni interface; (**b**) high-resolution magnification map of the interface between $M_{23}C_6$ and $\gamma$-Ni; (**c**) FFT of $M_{23}C_6$; (**d**) FFT of $\gamma$-Ni.

### 3.2. Microhardness

Figure 7 shows the cross-sectional microhardness distribution of coatings with different amounts of nano-WC added. It can be seen from Figure 7a that the hardness quickly increases from 350 $HV_{0.2}$ of the matrix to 500~700 $HV_{0.2}$ of the laser claddings. This means the dilution rates of coatings on the substrate made of 17-4PH martensitic stainless

steel are all very low. Moreover, the hardness curves in the coating section are relatively stable, indicating that the microstructure of the coating is relatively uniform at the macro level. Figure 7b is the average microhardness of the coatings with different contents. The microhardness of the coatings gradually increases with an increase in nano-WC content. For pure Ni60 coating (nano-WC weight percent is 0), the average hardness of the coating is measured at 506 $HV_{0.2}$. With a 5 wt.% increase in nano-WC content, the average hardness of the coating reaches 556 $HV_{0.2}$, representing a relative increase of 9.9%. Further increasing the nano-WC content to 10 wt.% results in an average hardness of the coating of 578 $HV_{0.2}$, showing a relative increase of 14.4%. A subsequent rise in nano-WC content to 15 wt.% leads to an average microhardness value for the coating of 593 $HV_{0.2}$, demonstrating a relative increase of approximately 17.3%. Finally, when nano-WC reaches a 20 wt.% content, the average microhardness of the coating reaches 643 $HV_{0.2}$, resulting in an increase of approximately 27.1%.

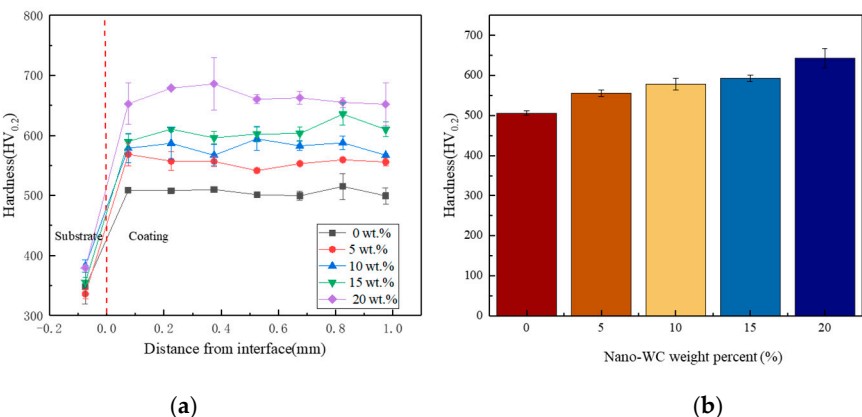

(**a**)  (**b**)

**Figure 7.** Microhardness of coatings with different contents: (**a**) the hardness distribution curves from the substrate to coating; (**b**) the average hardness of the coatings.

### 3.3. Tribological Properties

The friction and wear tests of coatings containing varying amounts of nano-WC were conducted at ambient temperature. The obtained friction coefficient curves are shown in Figure 8, in which it is evident that the friction coefficients of various coatings undergo a running-in stage of approximately 20 min, followed by a stable wear stage for all the tests. During the running-in stage, the contact between the grinding head and sample surface exhibited significant instability, resulting in substantial fluctuations in the friction coefficient. Subsequently, a relatively stable wear stage ensued with a consistent friction coefficient. The average friction coefficient of the stable wear stage is shown in Table 6. It can be found that the friction coefficient exhibits an increasing trend with the addition of nano-WC content. Notably, the pure Ni60 coating demonstrates the lowest friction coefficient at 0.296, while the 20 wt.% nano-WC-added composite coating exhibits the highest friction coefficient, reaching 0.348. The friction coefficient serves as a crucial indicator for evaluating the level of adhesion between the coating surface and the grinding material under dry sliding friction conditions. An increase in the friction coefficient enhances the adhesion wear resistance of the coating under such conditions. The main reason for the friction coefficient increasing is that there is a higher proportion of carbide particles in the composite coating, which increases the roughness of the friction surface. The findings of Qing Z et al. [20] align with this perspective, suggesting that the incorporation of composite phases in the coating can exert an influence on surface roughness. Overall, the friction coefficient of pure Ni60 and nano-WC+Ni60 is in the range of 0.30–0.35, indicating that the change in friction coefficient is not significant.

The wear rate of coatings with different nano-WC contents is shown in Figure 9. It can be found that the wear rate of the coating gradually decreases with the increase in nano-WC content. For the pure Ni60 coating, the wear rate is the highest, reaching

$2.12 \times 10^{-5}$ mm$^3$/(N·m). With 5 wt.%, 10 wt.%, 15 wt.%, and 20 wt.% nano-WC, this decreases to $1.50 \times 10^{-5}$ mm$^3$/(N·m), $7.37 \times 10^{-6}$ mm$^3$/(N·m), $\times 10^{-6}$ mm$^3$/(N·m), and $\times 10^{-6}$ mm$^3$/(N·m), respectively, which are 29.25%, 65.24%, 76.37%, and 85.19% lower than that of the pure Ni60 coating. The results indicate that the wear resistance of the coating has been significantly improved with nano-WC added. According to the microstructure analysis, it can be found that nano-WC is more likely to decompose, and the formation of $M_{23}C_6$-type carbides, which are rich in Cr and W, can be deduced. These carbides are widely distributed in the matrix and have higher hardness, thus reducing the rolling and cutting effect of the asperities on the surface of the grinding ball on the cladding coating.

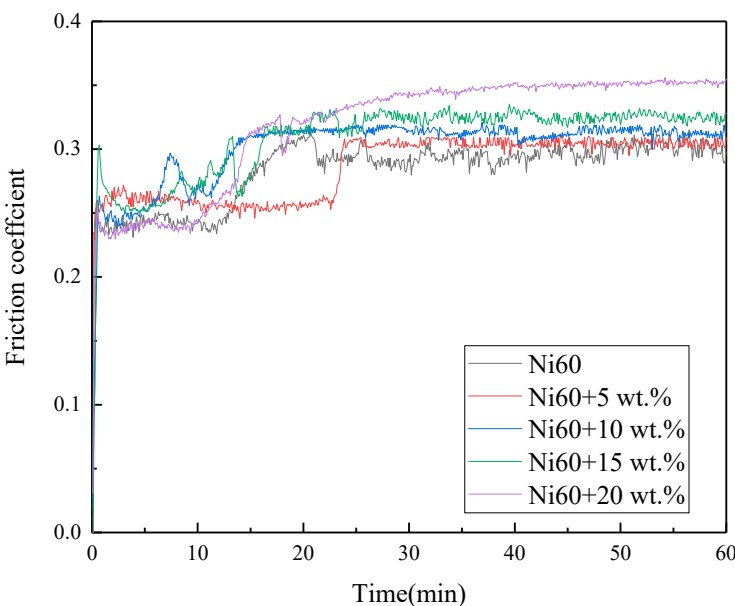

**Figure 8.** Friction coefficient curves of coatings with different content.

**Table 6.** The average friction coefficient of coatings with different nano-WC content.

| Nano-WC content (wt.%) | 0 | 5 | 10 | 15 | 20 |
|---|---|---|---|---|---|
| Friction coefficient | 0.296 | 0.305 | 0.313 | 0.324 | 0.348 |

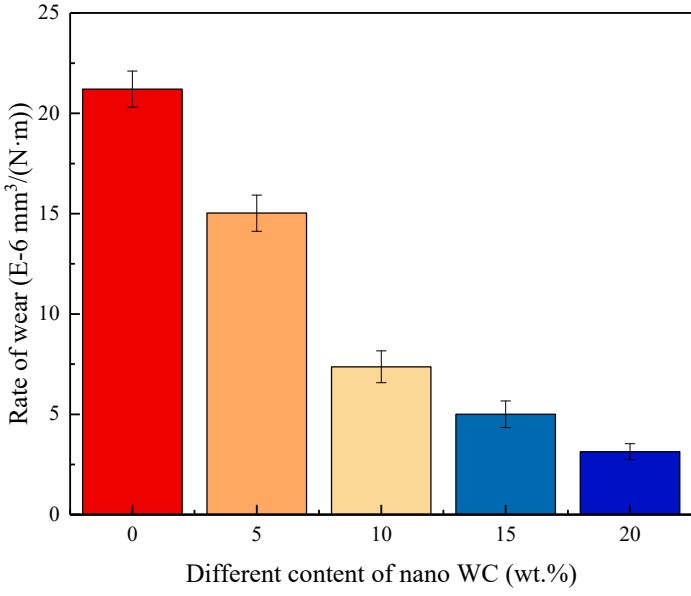

**Figure 9.** Volume wear rate of coatings with different contents.

## 4. Discussion

### 4.1. Role of Nano-WC in Microstructure Evolution

According to the above microstructure and texture characterization results, the mechanism of microstructure evolution after incorporating nano-WC can be deduced. As illustrated in Figure 10, the cladding of pure Ni60 is in a completely liquefied state in the laser melting pool, as Ni60 powder possesses self-fluxing properties and a low melting point. During the subsequent solidification process, highly oriented γ grains are initially formed. At the end of solidification, lamellar eutectic crystals consisting of γ and carbide emerge near γ grain boundaries and inter-dendritic zones due to the facile segregation of C and Cr atoms. As the temperature further decreases, a small quantity of borides are primarily composed of CrB precipitate adjacent to the lamellar eutectics. For the cladding of Ni60+nano-WC, although the theoretical melting point of WC is as high as 2870 °C, the actual melting point of nano-WC is much lower due to its nanoscale size. Nano-WC readily decomposes into free W and C atoms within the high-temperature molten pool. During this decomposition process, the remaining WC nanoparticles acts as heterogeneous nucleation sites for γ grains, resulting in a reduction in γ grain size and deterioration. The dissolved nano-WC generates abundant W and C atoms that rapidly attract Cr atoms in the liquid metal, leading to the formation of a compositional gradient region during solidification. This facilitates easier precipitation and growth of carbides during the subsequent cooling. Additionally, carbide precipitation and growth consume a significant amount of the Cr element, resulting in finer and more dispersed boride particle precipitation. Therefore, the resulting Ni60+nano-WC composite coating exhibits reduced grain size and weaker crystallographic orientation. The carbides transform from lamellar to rod or block morphology, while the borides become finer and more uniformly dispersed.

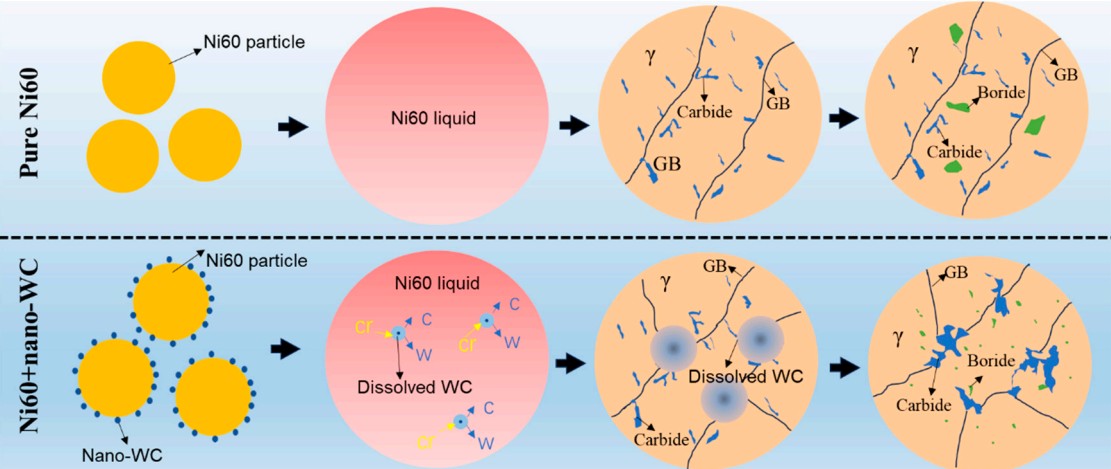

**Figure 10.** The schematic diagram illustrating the microstructure evolution of pure Ni60 and Ni60+nano-WC coating.

The microstructure is obviously different from that of the traditional micro-WC+Ni60 composite coating. In the micro-WC+Ni60 composite coating, despite the edges of WC particles exhibiting a certain degree of melting and dissolution, there remains a significant presence of undissolved WC particles within the tissue. Due to their high density, those micro-WC particles are prone to sinking to the bottom of the cladding layer, resulting in an uneven texture and performance; moreover, the addition of a significant quantity of micro-WC particles can more easily lead to the occurrence of coating cracking [13,14,21]. In contrast, nano-WC exhibits high specific surface energy, a low melting point, and a small scale, which facilitates its complete dissolution into the matrix. As a result, it leads to the formation of a more dispersed distribution of carbide and boride, enhancing uniformity in the microstructure and increasing crack resistance.

### 4.2. Hardening Mechanism

As confirmed by TEM, no significant amount of residual nano-WC was observed in the laser-cladded coatings, even with the addition of 20 wt.% nano-WC. This means that although the nano-WC particles can be fused and dissolved within the liquid molten pool, the resulting evolution of microstructures still enhances the coating's hardness. The reasons for this are as follows. Primarily, the precipitation of a substantial amount of $M_{23}C_6$ significantly improves the coating's hardness. These carbides are dispersed throughout the inter-dendritic zones, and form a strong network that acts as an effective barrier against dislocations, thereby greatly improving strength and hardness. Secondly, the smaller size of CrB particles and their dispersion contribute to enhancing the second-phase strengthening effects. The CrB material itself exhibits high material hardness, with a Vickers hardness ranging from 12 to 18 GPa. When CrB is present in the form of fine particles, it demonstrates an effective second-phase strengthening effect attributed to the Orowan mechanism. Specifically, during plastic deformation of the γ matrix, dislocation lines are unable to directly cut through the CrB particles, but can bend around them and eventually form a dislocation ring. The bending of dislocation lines leads to increased lattice distortion energy within the affected area, thus increasing the strengthening and hardness [22,23]. Additionally, the dissolution of W and C atoms resulting from nano-WC decomposition into the γ matrix reinforces the solid solution strengthening effects, consequently elevating the coating's hardness. In particular, W is an excellent solid solution strengthening element for γ. Pengfei Sun et al. found, with 12 wt.% added to Inconel 718, the microhardness was increased by about 110 HV [24]. Benefiting from the aforementioned hardening mechanisms, the 20 wt.% nano-WC+Ni60 coating exhibits exceptional hardness, reaching a value of 643 $HV_{0.2}$, which is comparable to the reported hardness (622 $HV_{0.2}$) of composite coatings containing 30% micron-WC + Ni60 by Ren M et al. [25].

### 4.3. The Wear Mechanism

To reveal the wear mechanism of the laser cladding coatings, the 3D topography and microstructures of the wear traces were observed by confocal laser scanning microscope and SEM, respectively. The results are shown in Figure 11. In particular, Figure 11a–c are the 3D topography and microstructures of the wear traces corresponding to the pure Ni60 coating, respectively. As seen in Figure 11a, the friction surface of the pure Ni60 coating exhibits a visible wear groove, and material uplift caused by plastic deformation is observed on both sides of the groove. According to Figure 11b,c, it can be found that there are a few fine scratches at the bottom of the groove, and the whole is relatively smooth. These characteristics of the wear trace indicate that the predominant wear mechanism is the two-body abrasive wear, where the so-called two body refers to the $Si_3N_4$ grinding ball and pure Ni60 coating. The reason for this lies in the fact that the friction pairs are constituted by $Si_3N_4$ grinding balls and pure Ni60 coatings. The grinding ball has a hard surface, while the pure Ni60 coating is softer and lacks internal hard particles. When two bodies underwent relative sliding motion under a specific pressure, the $Si_3N_4$ grinding ball penetrated the Ni60 surface and plowed the Ni60 to the sides of the contact surface as the slide started. In this stage, Ni60 was not removed from the surface but uplifted because of plastic deformation. As the slide continued, the grinding ball acted as the cutting tool and cut the Ni60 surface to form the obvious wear groove. Meanwhile, Ni60 chips were generated in front of the grinding ball and the material was removed from the surface, which led to significant volume wear of Ni60 coating. Because there are few hard particles in pure Ni60 coating, the bottom of the wear groove is relatively smooth.

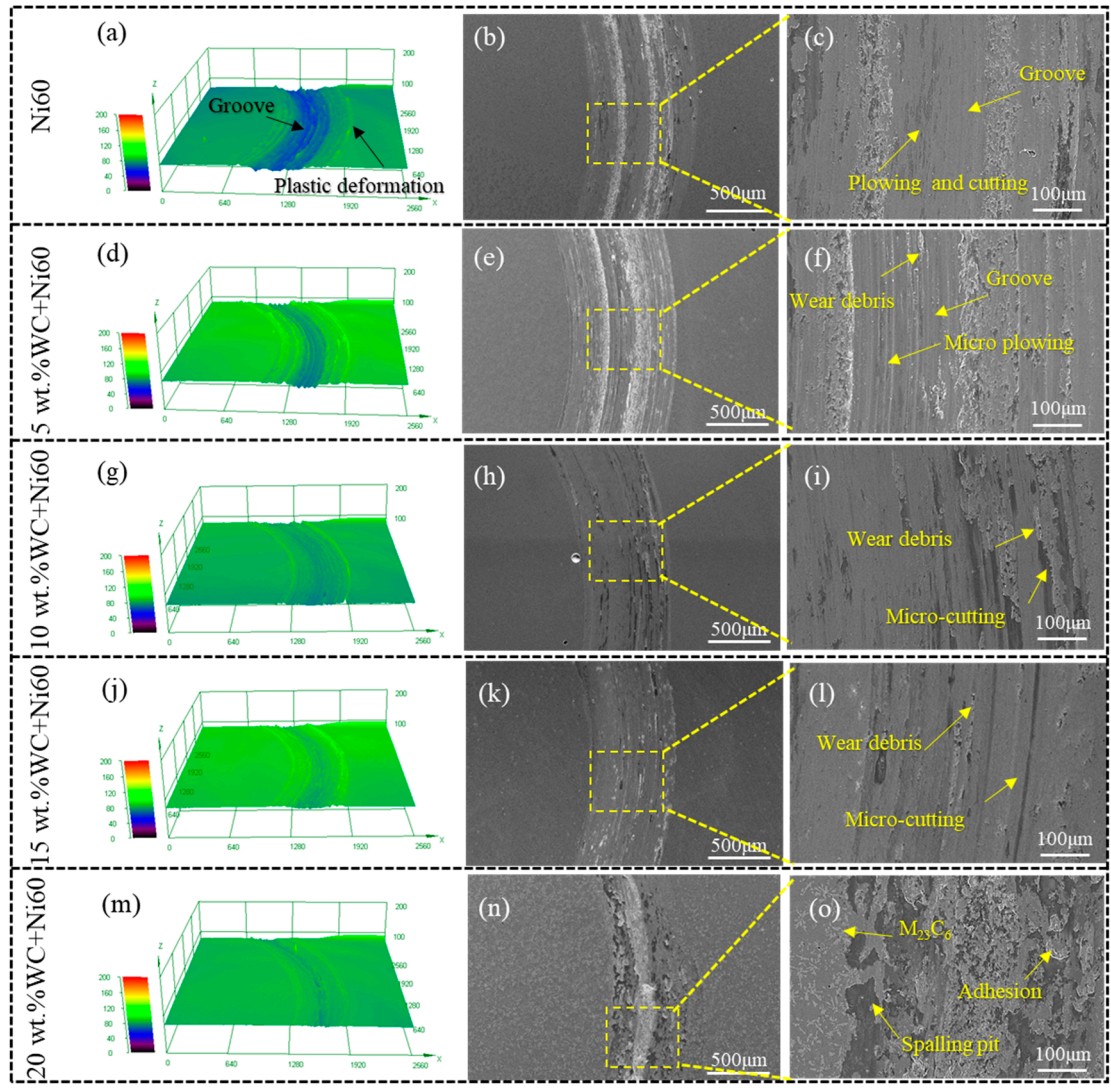

**Figure 11.** The 3D topography and microstructures of the wear traces with different nano-WC contents: (**a**–**c**) 0 wt.%; (**d**–**f**) 10 wt.%; (**g**–**i**) 20 wt.%; (**j**–**l**) 15 wt.%; (**m**–**o**) 20 wt.%.

With nano-WC added to the Ni60 coating, the wear traces gradually changed. As shown in Figure 11d–l, the depth and width of the wear groove caused by the grinding ball gradually decrease with the increase in nano-WC content, and there are a large number of micro-plowing and cutting traces at the bottom of the main wear groove. These facts demonstrate that the wear mechanism gradually changed from two-body abrasive wear to three-body abrasive wear. According to the microstructure analyses, the added body is the $M_{23}C_6$ particle indicated by nano-WC addition. These $M_{23}C_6$ particles separate from the Ni60 matrix and are trapped between the rubbing surfaces during friction, which leads to micro-plowing, cutting, and wear debris. Some of them can be re-inserted or cold-welded on the rubbing surface. Because the hardness of the coating enhanced with the nano-WC content increases, the penetration and cutting ability of the grinding ball is gradually weakened; therefore, the main wear groove tends to become shallow and narrow. As shown in Figure 11m–o, the 3D topography and microstructures of the wear traces changed obviously when the nano-WC addition increased to 20 wt.%, and there are significant spalling pits and adhesion blocks; moreover, these spalling pits are generally devoid of $M_{23}C_6$ particles. The wear trace morphology reveals that the main wear mechanism of the coating has changed from abrasive wear to adhesive wear. The reason for this is that the addition of 20 wt.% nano-WC significantly increased the coating's hardness,

making it difficult for the grinding ball to penetrate and effectively cut through the material. During rubbing of counter-bodies, the softer zones where $M_{23}C_6$ particles are barren can be cold-welded on the grinding ball and induce micro-junctions; these micro-junctions could rupture hereafter, resulting in the adhesion and spalling pits shown in Figure 11o. In general, the change in the wear mechanism greatly improves the wear resistance of the nano-WC/Ni60 composite coating.

According to Archard's wear theory, the wear quantity $V$ can calculated by

$$V = \frac{KNL}{H} \tag{2}$$

where $K$ is the friction coefficient, $N$ is the normal force, $L$ is the friction distance, and $H$ is the hardness of the friction surface [26]. As seen in Figures 7 and 8, the friction coefficient decreases with the increase in nano-WC addition, and all the stable friction coefficients are in the range of 0.30–0.35, indicating there are no obvious changes, whereas the hardness increases proportionally to the augmentation of nano-WC content addition. Therefore, it can be inferred that the main reason for the improvement in the wear resistance of the composite coating is the increase in coating hardness. The underlying hardening mechanisms are discussed in Section 4.2.

### 5. Conclusions

Ni60 and nano-WC composite coatings were prepared on the surface of 17-4PH martensitic stainless steel, and the microstructure, hardness, and wear mechanisms of the coatings were studied systematically. The main conclusions are as follows.

(1) Compared to the pure Ni60 coating, the Ni60+nano-WC composite coating exhibits a reduction in grain size and a decrease in crystallographic orientation strength. The lamellar $M_{23}C_6$ carbides undergo a transformation into a rod or block morphology, while the CrB borides become finer and more uniformly dispersed. This can be attributed to the dissolution of nano-WC during cladding.

(2) The microhardness of the coating exhibited a uniform increase upon the incorporation of nano-WC. For the coating containing 20 wt.% nano-WC, the microhardness reached 643.07 $HV_{0.2}$, exhibiting a remarkable increase of 27.12% compared to that of the pure Ni60 coating.

(3) The incorporation of nano-WC resulted in a significant enhancement of the coating's wear resistance. With an increase in nano-WC addition up to 20 wt.%, the wear mechanism gradually transitioned from two-body abrasive wear to three-body abrasive wear, and ultimately adherent wear.

**Author Contributions:** Conceptualization, X.Z., L.Q., T.L. and R.L.; Methodology, J.W., X.Z., Y.Z., M.R. and T.L.; Validation, J.W. and X.Z.; Investigation, M.R.; Data curation, J.W., Y.Z. and M.R.; Writing—original draft, J.W. and X.Z.; Writing—review & editing, J.W., X.Z., L.Q., Y.Z., T.L. and R.L.; Supervision, L.Q.; Project administration, R.L.; Funding acquisition, R.L. and X.Z. All authors have read and agreed to the published version of the manuscript.

**Funding:** This research was funded by the Natural Science Foundation of Jiangsu Province (Grants No BK20230671).

**Institutional Review Board Statement:** Not applicable.

**Informed Consent Statement:** Not applicable.

**Data Availability Statement:** Data are contained within the article.

**Conflicts of Interest:** The authors declare no conflicts of interest.

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
