# Peer review of "A Comprehensive Study on Microstructure and Wear Behavior of Nano-WC Reinforced Ni60 Laser Coating on 17-4PH Stainless Steel"

_coatings, doi:10.3390/coatings14040484_

Round 1

Reviewer 1 Report

Comments and Suggestions for Authors

The paper is written with quality and substance. The quality of the images is exceptional.

I have a couple of questions:
1. Did you use commercial nanoparticles or did you synthesize them in your lab? Something should be said about it in the article.
2. Is the chemical composition of the steel, which is shown in the text, given by the manufacturer or did you get the results from the analysis in your laboratory?
3. I couldn't tell from the text what was used as shielding gas? This should be explained in the text.

Reviewer 2 Report

Comments and Suggestions for Authors

The manuscript needs a comprehensive discussion section. A discussion is required in order to interpret the findings in the context of existing literature, provide insights into underlying mechanisms, and address potential implications and applications of the research.

The manuscript would benefit from comparing its results with those reported in previous studies on similar coatings. Such comparisons can highlight the novelty of the findings and provide a basis for further analysis.

The provided analysis of the wear traces with different nano-WC contents offers valuable insights into the evolving wear mechanisms and their relationship with nano-WC addition. Here are some critical points for further discussion and refinement:

The transition from two-body to three-body abrasive wear with increasing nano-WC content is well-described. Discuss how M23C6 particles, derived from nano-WC addition, contribute to this transition by becoming trapped between rubbing surfaces and inducing micro-plowing, cutting, and wear debris.

The observed changes in wear trace morphology, such as the decrease in depth and width of the wear groove with increasing nano-WC content, are attributed to the enhanced hardness of the coating. It would be beneficial to further discuss how the increased hardness affects the penetration and cutting ability of the grinding ball, leading to changes in wear trace morphology.

Discussing the role of M23C6 particles, particularly in barren zones where they are devoid of M23C6 particles, in inducing micro-junctions and subsequent adhesion and spalling pits would enhance the understanding of the wear behavior of the nano-WC/Ni60 composite coating.

It would be beneficial to delve deeper into the mechanisms underlying the influence of nano-WC content on friction behavior. Additionally, discussing how other factors, such as the morphology and distribution of carbide particles, may affect friction properties could provide a more comprehensive understanding.

Discuss how the distribution and morphology of carbide particles influence the wear mechanisms, such as by providing mechanical reinforcement or acting as lubricants.

Emphasize the role of M23C6 carbide precipitation in improving the coating's hardness. Discuss how the dispersion of carbides throughout the inter-dendritic zones contributes to strengthening mechanisms, such as barrier against dislocations and second-phase strengthening effects.

Discuss the contribution of smaller CrB particles and their dispersion in enhancing the second-phase strengthening effects. Explain how the finer and more uniformly dispersed CrB particles contribute to increased hardness through effective strengthening mechanisms.

Discuss how this reinforcement mechanism contributes to the overall increase in coating hardness with increasing nano-WC content.

Discuss the role of nano-WC nanoparticles as heterogeneous nucleation sites for γ grains. Elaborate on how the presence of nano-WC promotes nucleation and affects the grain size distribution in the solidified coating.

The calculation of interface mismatch and determination of strong interface strength between M23C6 and γ-Ni are significant findings. Explaining how this strong interface influences mechanical properties, such as crack propagation resistance and coating adhesion, would enhance the discussion.

The text lacks a clear structure, making it difficult to follow. Consider organizing the discussion into sections such as Introduction, Benefits of Laser Cladding, Role of WC Particles, Challenges with Nano-WC, Proposed Study, and Conclusion.

Reviewer 3 Report

Comments and Suggestions for Authors

Review on manuscript no coatings-2943997-peer-review-v1

Please find the comments for each section attached.

1. Title: it seems that the word steel is missing in the title

2. Materials and methods: authors mentioned that they have used 100x100x15mm3 substrate material for laser cladding. Was surface roughness investigated before cladding?

3. Chemical composition presented in Table 2 presents the content of elements for Ni60 powder, however, the results for the three elements are within some ranges. What was the specific chem comp of the powder then?

4. Authors mentioned “the nano-WC/Ni60 mixed powders with varying proportions were pre-prepared by electromagnetic stirring to ensure uniform mixing.” – please provide the process parameters.

5. Fig.1 – a, b, c, and d is missing in the figure caption

6. Methods, Table 4 – how these process parameters were selected? Were they optimized anyhow?

7. Microstructures: the authors have used description “Fig. 2 (a) and Fig. 2 (b) show the microstructure observed by SEM, Fig. 2 (b) is a partial enlarged view of Fig. 2 (a). Fig. 2 (a) and Fig. 2 (b) present the SEM images of the microstructure, with Fig. 2 (b) representing an enlarged view of Fig. 2 (a).  “ – please rewrite and shorten. There is too much repetitions which makes description hard to follow.

8. Figures 2,3 and 4 could be presented in one table to make the comparison clear. On such basis, it is easier to compare the changes in element distribution for different volume of WC.

9. Hardness results could be presented without decimal numbers.

10. Tribology: almost all materials exhibited the tendency of plateau after first minutes. It could be observed that the friction coefficient is stable and then increases again. What is the phenomenon behind it?

General comments: The reviewers find this paper attractive with interesting results presented. The microstructural analysis and tribological aspects of cladding coating are top-notch in terms of quality. Although the results are interesting, the paper lacks discussion. The authors should provide some comments on how their results fit into already-known literature. It is recommended to expand the results by the additional discussion in which the authors will relate their results to others. Try to compare the obtained cladding with similar ones found in the literature and compare if the process parameters used to produce them are enhancing/deteriorating their mechanical properties. Please try to highlight the importance of the proposed research and point the strong aspects of your research. Otherwise, the paper will be just a technical report with simple description of what was obtained.

Comments on the Quality of English Language

The additional proofreading is mandatory.

Round 2

Reviewer 2 Report

Comments and Suggestions for Authors

The submitted manuscript has been appropriately revised as needed. I believe it is now suitable for publication in its current form.

Reviewer 3 Report

Comments and Suggestions for Authors

The paper was improved thus it could be considered for publication. 

Comments on the Quality of English Language

Proofreading is recommended.